# Informed Consent in COVID-19-Research: An Ethical Analysis of Clinical Studies Performed during the Pandemic

**DOI:** 10.3390/healthcare11121793

**Published:** 2023-06-17

**Authors:** Katja Voit, Tobias Skuban-Eiseler, Marcin Orzechowski, Florian Steger

**Affiliations:** 1Institute of the History, Philosophy and Ethics of Medicine, Ulm University, 89081 Ulm, Germany; katja.voit@uni-ulm.de (K.V.); marcin.orzechowski@uni-ulm.de (M.O.); florian.steger@uni-ulm.de (F.S.); 2kbo-Isar-Amper-Klinikum Region München, 85540 Haar, Germany

**Keywords:** informed consent, COVID-19, clinical study, ethics

## Abstract

Health crises such as the current COVID-19 pandemic pose challenges to the conduct of clinical studies. Aspects of research ethics, such as obtaining informed consent (IC), can be complicated. We are concerned with whether or not the proper IC procedures were followed in the context of clinical studies at Ulm University in the years 2020 to 2022. We identified all protocols of clinical studies dealing with COVID-19 that the Research Ethics Committee of Ulm University has reviewed and voted on in the years 2020 to 2022. We then performed a thematic analysis regarding the following aspects: study type, handling of IC, type of patient information, means of communication, applied security precautions, and the approach to participants from vulnerable groups. We identified n = 98 studies that dealt with COVID-19. In n = 25 (25.51%), IC was obtained traditionally in written form, in n = 26 (26.53%) IC was waived, in n = 11 (11.22%) IC was obtained delayed, and in n = 19 (19.39%) IC was obtained by proxy. No study protocol was accepted that waived IC in case IC would have been required in times outside of pandemics. It is possible to obtain IC even in times of severe health crises. In the future, it is necessary to address in greater detail and with legal certainty which alternative methods of obtaining IC are possible and under which circumstances IC can be waived.

## 1. Introduction

Public health emergencies can pose challenges to medical research. Research may have to be conducted under very special considerations and challenges. These include safety precautions for research staff and study participants, the potentially very specific nature of the disease, particular time stress, and a potentially restricted availability of resources. Especially the recent pandemic of coronavirus disease 2019 (COVID-19) caused by severe acute respiratory syndrome coronavirus-2 (SARS-CoV-2) affected the conduct of clinical studies involving human beings. This was due to several reasons. Among these are: isolation or quarantine measures preventing patients from participating in the studies, limited access to study sites, or commitment of healthcare professionals to other tasks related to combatting the pandemic [1]. With respect to clinical research, these factors may significantly influence the process of informed consent (IC). This concerns both the content and quality of the information provided to the patient and the way consent is obtained. Within the European Union, clear guidelines exist regarding the performance of medical interventions on humans. Thus, no such intervention can be performed unless the patient’s free will is taken into account and an IC is obtained. This is necessary to preserve the human dignity and integrity of the person [2]. The Declaration of Helsinki is also clear about IC: medical research on humans is only possible if the subject is capable of giving consent and has been informed in detail about the intervention. Only when it has been ascertained that the subject has understood the information, is an IC to be obtained in written form or, if circumstances do not permit, in non-written form. In the latter case, consent must be formally documented and witnessed [3]. In Germany, these requirements are supported in particular by § 40b of the German Medicinal Products Act (AMG) [4]. Under ordinary circumstances, IC should be obtained without time pressure, in a calm environment, where the patient can ask any questions he or she may have. This way the patient can calmly decide whether he or she wishes to participate in a particular research project [5]. During the COVID-19 pandemic, some or all of these factors were difficult to establish. Measures of social distancing, for example, required research staff to communicate with study participants via electronic media. Reportedly, 57% of patient-investigator interactions took place remotely [6]. In addition, it was often not possible to obtain patient consent through traditional means, i.e., by signing a document, as this would have put research staff at increased risk of infection. Other ways, such as electronic signatures or verbal consents were used as an alternative [7]. In the wake of previous epidemics, such as the 2002–2004 SARS epidemic or the 2014–2016 Ebola outbreak, international initiatives have begun to develop ethical guidelines for conducting medical research during public health emergencies [8,9,10]. These documents agree that public health emergencies should not erode ethical standards, especially those under which IC is obtained. However, these documents also recognize that special circumstances may require that the process of IC be adapted [5]. In this situation, four factors play an important role. These can be described by four questions: when, who, if at all, and how [5]? First, consent can be delayed when participants are incapacitated to provide it, e.g., because they are unconscious or under sedation. Second, if the capacity to provide it is compromised, consent can be obtained not from the participants themselves but from proxies, which as a general rule, are legal representatives or legal guardians. Third, in particular situations, consent can be waived, especially if studies focus on retrospective data. Fourth, novel means of communication can be used in the patient information process and provision of consent. These can be dictated by security provisions in order to protect participants and researchers. However, important is that whatever modifications in obtaining of IC are adopted, they need to be harmonious with ethical guidelines for conduct of studies on humans and approved by research ethics committees.

This paper focuses on the process of IC as it was obtained during the COVID-19 pandemic from January 2020 to December 2022 in the context of medical studies whose protocols have been reviewed and voted on by the Research Ethics Committee (REC) at Ulm University. We addressed the following research questions:(a)Was IC obtained for all medical studies?In which situations was this not the case?(b)When was IC obtained? Are there studies where IC was obtained only after the research had started? Under which circumstances was this done?(c)From whom was IC obtained? Under which circumstances was IC sought by proxy?(d)In what way was IC obtained? Were there any special safeguards? Were other methods used, such as telecommunication, etc.?

## 2. Materials and Methods

We looked at the documentation of the electronic repository of the REC at Ulm University encompassing the time period from January 2020 to December 2022. Examined were all protocols submitted to the REC in this period of time in order to identify studies referring to research connected with COVID-19. The documents of the REC include a detailed description of the proposed research, the methods used in the research, methods and content of participant information, and templates for declaration of consent (DoC), which is to be signed by the participants, their legal guardians or representatives after a thorough information process (informed consent). The documentation also provides information on how participants were to be recruited and how the information required for IC was to be provided to them. In the evaluation, we have analyzed the last version of each particular study’s documents with supplements and amendments required by REC. If the applicant provided amendments to the documents after the REC’s vote with suggestions, we have evaluated this version of the application.

To provide an illustration of this information, descriptive statistics were performed on the content extracted from the applications. In the second step, the submitted applications for COVID-19 studies were qualitatively analyzed using the methods of documentary research and thematic analysis. Records of the REC need to meet four criteria of documentary research, which are authenticity, credibility, representativeness, and meaning [11]. Thematic analysis is a qualitative descriptive approach for the identification, analysis, and reporting of common patterns or themes that extend throughout a set of analyzed material [12,13]. For deductive categorization, the recommendations of the European Medicines Agency (EMA) [10,14], guidance from the Working Group on Medical Ethics Committees of the Federal Republic of Germany e.V. (AKEK) [15], and results of a first survey of German ethics committees in spring 2020 [16] were used. According to these, challenges to IC during the COVID-19 pandemic were particularly present in the following aspects: (i) IC may be waived; (ii) IC is obtained late, (iii) IC is obtained by proxy, (iv) design of IC has to be applied to the special situation of the pandemic. We used the aforementioned aspects as categories for our analysis. The thematic analysis of the full text of documents pertinent to the research questions was conducted. Specific statements from the documents were sought and coded manually in order to identify the data that matched the categories. From the identified study applications, we have extracted the following information: study type (intervention, observational), whether IC will be sought, reasons for waiving IC, type of patient information documents, means of communication of information, applied security precautions, way of recording of IC, whether IC will be delayed and reasons for this, whether IC will be provided by proxy and reasons for this, approach to participants from vulnerable groups, and other notable features of the proposed studies. The coding process involved highlighting relevant information with notes on the text. This information was then extracted, assigned to particular categories, and grouped in thematic tables.

## 3. Results

In the period under review, we identified n = 1529 applications for new studies submitted for the vote of the REC and documented in the REC’s electronic repository. This number encompasses aggregated applications, which in some cases enclose amendments. Of the n = 1529 aggregated applications, n = 98 (6.41%) were studies related to COVID-19. In the year 2020, n = 45 applications were submitted for studies related to COVID-19 (8.72% of all applications in this year); in 2021, n = 35 applications were submitted (6.47%); and in 2022, n = 18 applications were submitted (3.81%) (Figure 1). Of all n = 98 studies related to COVID-19 we could identify in the time period from 2020 to 2022, n = 10 were interventional studies (10.2%), and n = 88 were observational studies (89.8%). Among the observational studies, n = 25 (28.4%) were retrospective studies, n = 11 (12.5%) were online surveys, and n = 5 (5.68%) were surveys conducted via post. Of n = 10 intervention studies, n = 9 were studied under the German Medicinal Products Act (AMG) [4]. One study was an intervention study in which an open-label placebo and breathing exercises were provided. n = 7 intervention studies (70.0%) were multi-center studies, and n = 3 were mono-center studies (30.0%) (Table 1).

### 3.1. Waiver of IC

IC was to be obtained in n = 72 (73.47%) submitted studies. In n = 26 studies (26.53%) no IC was to be obtained. With one exception, all of these studies (n = 25) were retrospective studies, which required no IC if the data was anonymized. One study did not foresee obtaining IC; however, in this study, no health data or data on sexuality was collected. Therefore, this study was not considered to be relevant for consultation by the REC and no vote of the REC was issued. One study protocol foresaw waiver of IC due to security precautions, i.e., increased risk of infection during the process of informing the participants. After a negative opinion of the REC, the study protocol was modified and the application was re-submitted as a retrospective study, for which IC was not necessary.

### 3.2. Delayed Informed Consent

In n = 11 (11.22%) of all n = 98 studies submitted, the documents were considered delayed IC. This measure was considered for participants unable to provide IC personally because they were unconscious, under respiration, or sedated. In summary, a delayed IC was used particularly when the subjects were intensive care patients or in emergency situations in which the potential study participants were temporarily unable to consent and for whom it was not possible to obtain an IC in advance. In such situations, the study protocols foresaw that consent should be obtained after the decision to include the subject in the study. In n = 9 submitted studies, study protocols foresaw the inclusion of legal representatives as proxies who can give consent. In two observational studies involving an investigation of bodily materials and without any additional risks for participants, the inclusion of legal representatives was not envisaged. In these cases, participants could provide their consent or withdraw from the study after regaining consciousness. A special situation presented cases of emergency treatment, when the participants could not provide consent themselves and legal representatives were not available or unable to give consent for inclusion in the study, e.g., because of emotional overburden. In such situations, n = 3 study protocols–two interventional and one observational study–foresaw the inclusion of an independent physician not involved in the study. The physician should confirm the emergency treatment indication and the inclusion of the patient in the study.

In one of these three cases, the study protocol foresaw that the independent physician should interview related parties, i.e., relatives or other persons with a relationship to the patient. The aim of the interview should be to ascertain whether participation in the trial represents the alleged patient’s will. It should be determined whether the patient expressed his/her will regarding inclusion in a therapeutic clinical trial in an oral or written statement before the onset of acute illness. In these cases, the related party should sign the IC documentation of the evaluation of the alleged patient will. In cases when the alleged will not be ascertained or no related party could be identified, inclusion in the trial was not possible.

### 3.3. IC by Proxy

In addition to n = 9 studies that foresaw the inclusion of legal participants for obtaining IC in case of unconscious or sedated patients, n = 10 (10.2%) study protocols foresaw informed consent by proxy. IC by proxy was used when study participants were unable to provide consent. This was the case, for example, if the study participants were suffering from dementia or if they were minors who were legally unable to give IC. In n = 8 studies involving minor participants, the IC should be provided by parents or legal guardians. N = 6 studies foresaw the assent of minor participants after the provision of age-appropriate information. In n = 2 studies, in which elderly over 65 years were included, study protocols were provided for IC by proxy if the prospective participants could not give IC themselves due to permanent illness (Table 2).

### 3.4. Information Process for Obtaining IC

The majority of the studies (n = 56, 57.14%) foresaw both a verbal and written information process. Prospective participants in the studies were to receive written information material. The verbal information process was to be conducted personally by a physician participating in the study. n = 25 (25.51%) retrospective studies foresaw no information process due to the fact that IC was not necessary to obtain in these studies. In n = 11 (11.22%) study applications that consisted of online questionnaires, only written information was to be provided to participants. Two such online studies provided the possibility to ask questions prior to consent, either via email or phone. n = 5 (5.1%) studies envisaged questionnaires sent via post. Furthermore, in these cases, only written information was provided with the possibility to ask questions via phone. One study that did not involve research on humans foresaw no information process (Figure 2).

### 3.5. Security Precautions

The majority of the studies did not foresee any security precautions during the process of informing the participants or obtaining the IC. Any precautions that prevented direct contact between the research team and participants ensued from the design of the study, which was to be conducted either by electronic means (online surveys) or via post or phone. Submitted study protocols did not mention any measures for the protection of the researchers or participants, e.g., masks or protective suits. Obtaining electronic IC was not planned in any of the analyzed studies if it was not required by the study design, i.e., online surveys.

## 4. Discussion

The evaluation of IC in clinical studies submitted to the REC of Ulm University clearly shows that it is possible to obtain IC during a pandemic. In all studies we analyzed in which obtainment of IC was required, IC was received. We could not identify a study in which IC was waived. The REC of Ulm University could build on approaches to obtaining IC in comparable situations, which took place even before the COVID 19-pandemic [17]. Furthermore, there are and have been constellations outside of pandemics in which it is not possible to obtain written IC from a study participant and in which alternative solutions must be found. Examples include studies with unconscious patients or children who are unable to give formal consent [18,19]. 

As the obtainment of IC is ethically central to clinical studies, the Declaration of Helsinki explicitly addresses situations in which it might be challenging to obtain IC: In the event that consent cannot be expressed in written form, the Declaration of Helsinki stipulates that non-written consent must be formally documented and witnessed [3]. 

There exist several possibilities to obtain IC in such challenging cases. This is of paramount importance as it may be critical to conduct research projects that aim to end the health emergency. However, even during a pandemic that threatens the health of many people such as the COVID-19 pandemic, the IC process must not be abandoned as it is one of the most important pillars of research ethics [20]. Alternative methods to obtain IC must be critically reviewed for their suitability, as they may prove to be ethically questionable [21]. It might be helpful to combine several methods of obtaining IC [5,20,22]. However, this might require specific training for the study teams, who are usually used to collecting IC in the traditional way, i.e., in written form [22]. In some cases, too much emphasis is and has been placed on obtaining a signature or any other way of consent to the IC document than on the very process of informing the potential participant about the study at hand [22,23]. This information process can take many forms, such as online surveys [7] or other electronic ways (eIC) [24]. It must be ensured that an eIC process is technically secure, access to patient-relevant data is limited by the eIC system itself, and the identity of the study participants is protected [25]. Furthermore, the information must be encrypted and stored appropriately so that it can be retrieved quickly [26]. In this way, eIC can even increase patient usability, satisfaction, knowledge, and trust scores compared to traditional paper consent [27]. In very rare cases, it may also be ethically opportune to waive IC. It is not uncommon, however, to be unable to predict in which studies this might be possible under the conditions of a particular pandemic, which is why there is a desideratum for regulatory standards for such IC exceptions [28].

Questions regarding the requirement versus waiver of IC represent one of the greatest challenges in medical ethics [16,20,28,29]. Largent et al. (2021) note that in order to promote research, it is conceivable to waive the traditional obtaining of an IC in the event of a pandemic. This traditional form is subsumed by the authors to include not only signing a document but also videoconferencing and electronic signatures. If obtaining an IC that is traditional in this sense would result in a study becoming impractical, a waiver of IC might be warranted, even if such an exception would not occur in non-pandemic times. [28]. However, deciding on the degree of risk and, hence, whether a study qualifies for a waiver is extremely difficult during a pandemic. In this regard, the handling of IC in some studies conducted with COVID-19 patients was ethically unacceptable [21,28]. It should not go unmentioned that, in addition to ethical concerns, there are also clear legal requirements in Germany. According to these, it is a breach of law to conduct a study without obtaining an IC when one is required. The approach of the REC of Ulm University was very cautious in this sense as it did not alter its assessment regarding the necessity to obtain IC compared to situations outside a pandemic. It did not give a positive ethical vote for studies not including the obtainment of IC when this would have been required outside of a pandemic situation. If the gold standard envisaged in the Declaration of Helsinki, i.e., voluntary IC by the subject, preferably in written form [3], was not possible, alternatives had to be used. Alternatives accepted by the REC of Ulm University were IC by proxy (n = 10, 10.2%), delayed IC (n = 2, 2.04%), and a combination of delayed IC and IC by proxy (n = 9, 9.18%).

In its handling of IC, the approach of the REC of Ulm University corresponds to those of other ethics committees in Germany. In a status-quo survey conducted in April 2020 among all 52 medical ethics committees in Germany that are established according to state law, challenges and possible ways of dealing with IC during the pandemic were documented. In particular, (1) the question of the ability to include patients who are incapable of giving IC, (2) criteria for a lack of ability to give IC, and (3) barriers due to isolation measures posed the greatest problems [16]. Research ethics issues in this context were also addressed during the summer meeting of the working group of medical ethics committees in the Federal Republic of Germany in June 2021: In addition to the difficulties already mentioned, potential study participants were often regarded as severely limited in their health, or their surrogates are difficult to reach or the overall validity of IC might furthermore be challenged by stress and fear experienced by the participant [30]. In the solutions proposed, the importance of written IC once more was emphasized, but alternatives such as IC via telephone were also considered, especially when isolation measures were necessary. In the case of those incapable of giving IC or those with limited capacity to do so, a combination of IC and IC by proxy was suggested [16]. The handling of IC in the clinical studies submitted to the REC of Ulm University is thus consistent with the approaches practiced or considered throughout Germany. In Ulm, too, IC by proxy by authorized relatives, subsequent IC, or IC by means other than written consent or a combination of several variants was used. 

So far there are no binding legal requirements for the conduct of clinical trials during the COVID-19 pandemic. However, published guidelines recommend obtaining informed consent despite any adversity, such as the isolation of potential study participants [8,10,14,15]. Ulm University REC’s approach to IC during the COVID-19 pandemic is consistent with national and international recommendations. The REC of Ulm University thus shares the majority opinion of international medical ethicists, according to which no exceptions may be made in scientific research even during a pandemic [31,32,33], and special attention should be paid to ethical aspects [34,35]. 

Questions related to IC arise worldwide during the pandemic. From January 2020 to December 2022, 757 studies can be found in EudraCT for the search term “COVID-19” alone [36]. Within the EU, there is a common legal basis for dealing with IC through legal documents such as Directive 2001/20/EC [37], Regulation (EU) No 536/2014 [2], Regulation (EU) 2016/679 [38] and, albeit non-binding, the EMA’s recommendations on clinical trials during the pandemic [10]. The REC of the Ulm University approach is in line with EU law and the EMA recommendations. 

Research and recommendations on IC during the pandemic were also made in the USA, China, and South Africa, among others. Many challenges that have been stated were comparable to those mentioned above [5,8,22,23,39]. Among other things, it was mentioned that it may be difficult to obtain IC by proxy when quick decisions are needed and that alternative technical solutions can be helpful. At the same time, however, it was pointed out that these are not problems that only occur during the COVID-19 pandemic, but are already known from similar emergency health situations in the past and can also be expected in the future [22,23]. A waiver of IC is consistently regarded as being possible only in very rare cases and only within the framework of an individual case decision after consultation and agreement with the medical ethics committee [5]. Clear criteria as to when an IC may be waived have hardly been addressed in international recommendations so far [40]. 

To promote medical progress and to react to health emergencies, it is of high importance to conduct clinical trials. This became particularly clear during the COVID-19 pandemic. Without appropriate research, it is highly unlikely that this health crisis could have been overcome in such a relatively short time. Nevertheless, basic ethical requirements for conducting studies, such as obtaining IC, must not be disregarded even in such challenging situations. The analysis we conducted shows that this is possible and is being carried out. This is a reassuring finding, however, in the future, it will be necessary to address in even greater detail and with legal certainty which alternative methods of obtaining IC are possible and under which rare circumstances IC can be waived.

## 5. Conclusions

Our study shows that even in times of severe health crises it is not only important but also possible to observe ethically relevant pillars in the conduct of clinical studies such as obtaining an IC. Since conducting clinical studies in such situations can be critical to ending the health crisis, and since the process of informing participants and obtaining IC can be complicated in different ways depending on the nature of the disease, research centers should establish alternative and legally sound methods for obtaining IC. Obtaining IC is not only a legal necessity. The provision of detailed information, the possibility to ask questions, and the prohibition to conduct a study without the consent of a subject is an expression of respect for the human dignity and integrity of each person. Thus, waiving IC must remain a rarity and is reserved for particular cases, which must be assessed individually by the ethics committees. Medical research has a high responsibility to contribute to ending health crises. At the same time, it must be aware of its ethical obligations at all times. Thus, it is indispensable to ensure the continuation of research that is impeccable in terms of research ethics.

## Figures and Tables

**Figure 1 healthcare-11-01793-f001:**
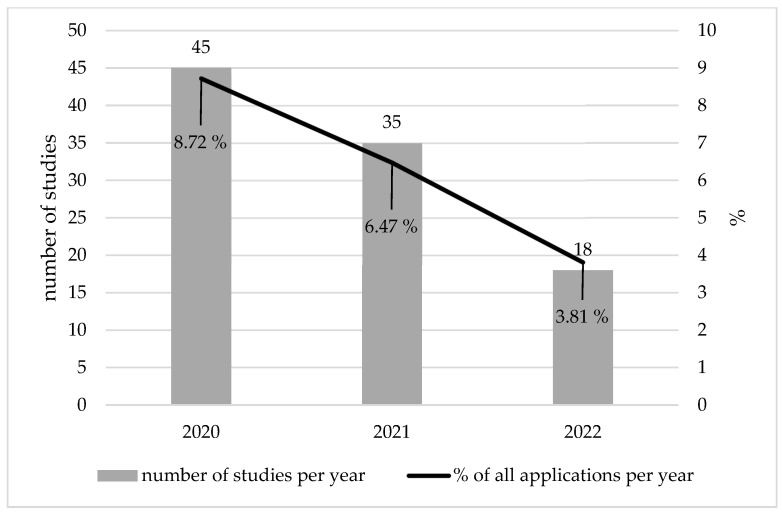
Number of COVID-19-related studies and their percentage of all studies submitted to the REC of Ulm University in the years 2020–2022.

**Figure 2 healthcare-11-01793-f002:**
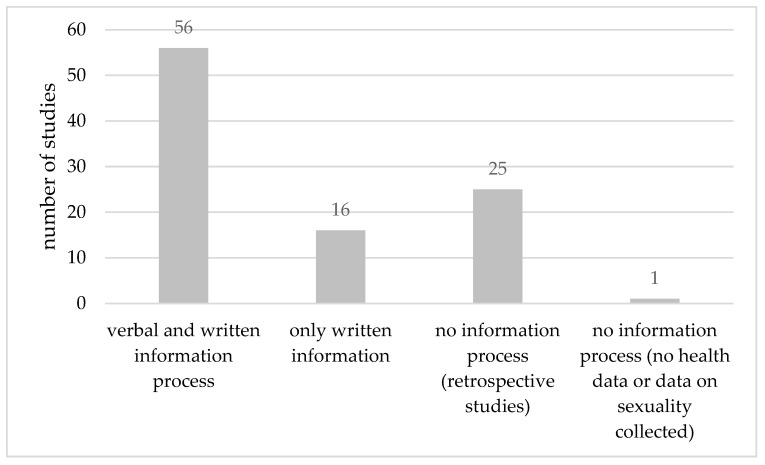
Handling of information process in all n = 98 COVID-19-related studies.

**Table 1 healthcare-11-01793-t001:** Characteristics of the studies identified in the documentation of the Research Ethics Committee (REC) at Ulm University (years 2020–2022).

**total number of identified studies**	n = 1529
**number of studies related to COVID-19**	n = 98 (6.41% of all studies)
**observational studies:** retrospective studies online surveys surveys conducted via post	n = 88 (89.8% of COVID-19-related studies) n = 25 n = 11 n = 5
**interventional studies:** studies under the German Medicinal Products Act (AMG) study involving open-label placebo and breathing exercises multi-center studies mono-center studies	n = 10 (10.2% of COVID-19-related studies) n = 9 n = 1 n = 7 n = 3

**Table 2 healthcare-11-01793-t002:** Handling of IC in n = 72 COVID-19-related studies in which IC was to be obtained.

Total number of identified studies related to COVID-19	n = 98 (100%)
**No vote of REC required (no health data or data on sexuality was collected)**	**n = 1 (1.02%)**
**Waiver of IC (retrospective studies)**	**n = 25 (25.51%)**
**IC obtained**	**n = 72 (73.47%)**
IC by the study participant	n = 51 (70.83%)
Delayed IC, of which…	n = 11 (11.22%)
inclusion of legal representatives as proxies	n = 9
no inclusion of legal representatives as proxies	n = 2
IC by proxy, of which…	n = 10 (10.2%)
studies involving minor participants	n = 8
studies involving elderly participants	n = 2

## Data Availability

Restrictions apply to the availability of these data. Data was obtained from the Research Ethics Committee of the Ulm University and are available from the authors with the permission of the Research Ethics Committee of the Ulm University.

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
