# Peer review of "Informed Consent in COVID-19-Research: An Ethical Analysis of Clinical Studies Performed during the Pandemic"

_healthcare, 2023, doi:10.3390/healthcare11121793_

Round 1
Reviewer 1 Report
In the introduction, explaining the German law regarding informed consent in medical treatment could be interesting. also analysing other nations' laws regarding informed consent and its legal and ethical issues.
In what kind of studies was delayed consent or consent by proxy used?
In which cases a waiver of informed consent is acceptable?
The conclusion must be enriched.
Author Response
Dear Reviewer,
thank you very much for your helpful comments. They helped us in reviewing and carefully revising our manuscript. We have considered each suggestion carefully and made changes in our manuscript accordingly. Below you will find our responses to all suggestions and comments. All changes to the manuscript have been made in the tracked changes mode.
Looking forward to hearing from you, with best regards.
Tobias Skuban-Eiseler
Corresponding author
(on behalf of all authors)
Your comments:
Comment 1:
In the introduction, explaining the German law regarding informed consent in medical treatment could be interesting. also analysing other nations' laws regarding informed consent and its legal and ethical issues.
Reply 1:
Thank you very much for your comment, which we were happy to work on. We have now supplemented the introduction and refer to the Declaration of Helsinki and European regulations. We also mention the German Products Act, which supports the aforementioned regulations.
The corresponding insertion reads (lines 42-54):
Within the European Union, clear guidelines exist regarding the performance of medical interventions on humans. Thus, no such intervention can be performed unless the patient's free will is taken into account and an IC is obtained. This is necessary to preserve human dignity and integrity of the person [2]. The Declaration of Helsinki is also clear about IC: medical research on humans is only possible if the subject is capable of giving consent and has been informed in detail about the intervention. Only when it has been ascertained that the subject has understood the information, is an IC to be obtained in written form or, if circumstances do not permit, in non-written form. In the latter case, consent must be formally documented and witnessed [3]. In Germany, these requirements are supported in particular by § 40b of the German Medicinal Products Act (AMG) [4].
Comment 2:
In what kind of studies was delayed consent or consent by proxy used?
Reply 2:
We have supplemented sections 3.2 and 3.3. accordingly.
The insertion regarding delayed IC reads (lines 214-217):
In summary, a delayed IC was used particularly when the subjects were intensive care patients or in emergency situations in which the potential study participants were temporarily unable to consent and for whom it was not possible to obtain an IC in advance.
The insertion regarding IC by proxy reads (lines 243-245):
IC by proxy was used when study participants were unable to provide consent. This was the case, for example, if the study participants were suffering from dementia or if they were minors who were legally unable to give IC.
Comment 3:
In which cases a waiver of informed consent is acceptable?
Reply 3:
We have added to the discussion and have described the circumstances under which a waiver of IC is conceivable.
The corresponding section in the text now reads (289 - 301):
Questions regarding the requirement versus waiver of IC represent one of the greatest challenges in medical ethics [16,20,28,29]. Largent et al. (2021) note that in order to promote research, it is conceivable to waive the traditional obtaining of an IC in the event of a pandemic. This traditional form is subsumed by the authors to include not only signing a document, but also videoconferencing and electronic signatures. If obtaining an IC that is traditional in this sense would result in a study becoming impractical, a waiver of IC might be warranted, even if such an exception would not occur in non-pandemic times. [28]. However, deciding on the degree of risk and, hence, whether a study qualifies for a waiver is extremely difficult during a pandemic. In this regard, the handling of IC in some studies done with COVID-19-patients was ethically unacceptable [21,28]. It should not go unmentioned that, in addition to ethical concerns, there are also clear legal requirements in Germany. According to these, it is a breach of law to conduct a study without obtaining an IC when one is required.
Comment 4:
The conclusion must be enriched.
Reply 4:
We have expanded the Conclusions and, in this context, have emphasized ethical considerations in particular even more clearly.
It now reads (lines 521 – 534):
Our study shows that even in times of severe health crises it is not only important but also possible to observe ethically relevant pillars in the conduct of clinical studies such as obtaining an IC. Since conducting clinical studies in such situations can be critical to ending the health crisis, and since the process of informing participants and obtaining IC can be complicated in different ways depending on the nature of the disease, research centers should establish alternative and legally sound methods for obtaining IC. Obtaining IC is not only a legal necessity. The provision of detailed information, the possibility to ask questions and the prohibition to conduct a study without the consent of a subject is an expression of respect for the human dignity and integrity of each person. Thus, waiving IC must remain a rarity and is reserved for particular cases, which must be assessed individually by the ethics committees. Medical research has a high responsibility to contribute to ending health crises. At the same time, it must be aware of its ethical obligations at all times. Thus, it is indispensable to ensure the continuation of research that is impeccable in terms of research ethics.
Reviewer 2 Report
This is a paper on a topic of great importance given the limitations researchers were under during the COVID pandemic. Examining informed consent adherence by the research staff of projects involving COVID patients provides a valuable measure of the strengths and weaknesses of current approaches. The authors focus on one hospital, however, and this limits the findings which may or may not fit other areas in a country. To be fair, there was an attempt to relate the Ulm study to approaches to informed consent in the rest of Germany. Since Germany's central government controls matters of informed consent, it is safer to postulate that the nation as a whole, should reflect what was done at Ulm, especially in its conservative approach to retaining the strict rules for informed consent even in the midst of the pandemic.
It would be interesting to read more about potential exceptions to informed consent and reasons that could be given for such exceptions. What situation might warrant suspension of informed consent? Another question is whether, in the stresses of a pandemic, utilitarian considerations focusing on moderating and halting the pandemic make it easy to justify what otherwise would be an egregious violation of informed consent. It becomes tempting to "triage" certain ethical considerations in the name of saving lives. Prudence is the necessary virtue in these cases, the ability to pick out the best course of action in a particular situation. Good facts are the key to good ethics in situations decided on a case-by-case basis, and among the tasks of the ethics committee may be to sort facts from falsehood in order to make the most prudent determination of whether an ethical suspension of informed consent is possible.
Overall, the quality of English is good. Sometimes the language is awkward, as in sentence three of the Introduction. Re-reading could focus on whether any material in the discussion or conclusion sections belong in the introduction, perhaps a more extended discussion of protocols in the United States and European nations outside Germany. Re-read carefully for clarity and organization.
Author Response
Dear Reviewer,
thank you very much for your helpful comments. They helped us in reviewing and carefully revising our manuscript. We have considered each suggestion carefully and made changes in our manuscript accordingly. Below you will find our responses to all suggestions and comments. All changes to the manuscript have been made in the tracked changes mode.
Looking forward to hearing from you, with best regards.
Tobias Skuban-Eiseler
Corresponding author
(on behalf of all authors)
Your comments:
Comment 1:
This is a paper on a topic of great importance given the limitations researchers were under during the COVID pandemic. Examining informed consent adherence by the research staff of projects involving COVID patients provides a valuable measure of the strengths and weaknesses of current approaches. The authors focus on one hospital, however, and this limits the findings which may or may not fit other areas in a country. To be fair, there was an attempt to relate the Ulm study to approaches to informed consent in the rest of Germany. Since Germany's central government controls matters of informed consent, it is safer to postulate that the nation as a whole, should reflect what was done at Ulm, especially in its conservative approach to retaining the strict rules for informed consent even in the midst of the pandemic.
Reply 1:
Thank you very much for this comment. You are right. There are national regulations regarding informed consent in Germany. However, the more than 50 ethics committees in Germany interpret these regulations very differently in practice. There is no overarching or central institution that has the authority to enforce directives in this regard. This is certainly a very important aspect and we thank you for pointing it out, but we believe that this, admittedly important, issue is outside the focus of our article.
Comment 2:
It would be interesting to read more about potential exceptions to informed consent and reasons that could be given for such exceptions. What situation might warrant suspension of informed consent?
Reply 2:
We have taken up this aspect and extended the discussion accordingly.
The corresponding section in the text now reads (289 - 301):
Questions regarding the requirement versus waiver of IC represent one of the greatest challenges in medical ethics [16,20,28,29]. Largent et al. (2021) note that in order to promote research, it is conceivable to waive the traditional obtaining of an IC in the event of a pandemic. This traditional form is subsumed by the authors to include not only signing a document, but also videoconferencing and electronic signatures. If obtaining an IC that is traditional in this sense would result in a study becoming impractical, a waiver of IC might be warranted, even if such an exception would not occur in non-pandemic times. [28]. However, deciding on the degree of risk and, hence, whether a study qualifies for a waiver is extremely difficult during a pandemic. In this regard, the handling of IC in some studies done with COVID-19-patients was ethically unacceptable [21,28]. It should not go unmentioned that, in addition to ethical concerns, there are also clear legal requirements in Germany. According to these, it is a breach of law to conduct a study without obtaining an IC when one is required.
Comment 3:
Another question is whether, in the stresses of a pandemic, utilitarian considerations focusing on moderating and halting the pandemic make it easy to justify what otherwise would be an egregious violation of informed consent. It becomes tempting to "triage" certain ethical considerations in the name of saving lives. Prudence is the necessary virtue in these cases, the ability to pick out the best course of action in a particular situation. Good facts are the key to good ethics in situations decided on a case-by-case basis, and among the tasks of the ethics committee may be to sort facts from falsehood in order to make the most prudent determination of whether an ethical suspension of informed consent is possible.
Reply 3:
Thank you very much for this comment. It is precisely the recourse to utilitarian arguments that bears the risk of not taking into account certain ethical pillars of research. From a purely utilitarian point of view, such situations, as you correctly point out, boil down to trade-offs. In the context of such trade-offs, certain ethical aspects would then have to be protected separately, which would be alien to a purely utilitarian argument. You rightly point out that virtue ethics can be very helpful here.
Comment 4:
Comments on the Quality of English Language
Overall, the quality of English is good. Sometimes the language is awkward, as in sentence three of the Introduction. Re-reading could focus on whether any material in the discussion or conclusion sections belong in the introduction, perhaps a more extended discussion of protocols in the United States and European nations outside Germany. Re-read carefully for clarity and organization.
Reply 4:
We had the manuscript carefully proofread by a native speaker and improved the language accordingly.
Round 2
Reviewer 1 Report
ok